# FRN: Fractal-Based Recursive Spectral Reconstruction Network

**Ge Meng**[1], **Zhongnan Cai**[1], **Ruizhe Chen**[1], **Jingyan Tu**[1], **Yingying Wang**[1],
**Yue Huang**[1], **Xinghao Ding**[1,*]

[1]Key Laboratory of Multimedia Trusted Perception and Efficient Computing,
Ministry of Education of China, Xiamen University, Xiamen, Fujian, China
mengge0001@gmail.com

## Abstract

Generating hyperspectral images (HSIs) from RGB images through spectral reconstruction can significantly reduce the cost of HSI acquisition. In this paper, we propose a Fractal-Based Recursive Spectral Reconstruction Network (FRN), which differs from existing paradigms that attempt to directly integrate the full-spectrum information from the R, G, and B channels in a one-shot manner. Instead, it treats spectral reconstruction as a progressive process, predicting from broad to narrow bands or employing a coarse-to-fine approach for predicting the next wavelength. Inspired by fractals in mathematics, FRN establishes a novel spectral reconstruction paradigm by recursively invoking an atomic reconstruction module. In each invocation, only the spectral information from neighboring bands is used to provide clues for the generation of the image at the next wavelength, which follows the low-rank property of spectral data. Moreover, we design a band-aware state space model that employs a pixel-differentiated scanning strategy at different stages of the generation process, further suppressing interference from low-correlation regions caused by reflectance differences. Through extensive experimentation across different datasets, FRN achieves superior reconstruction performance compared to state-of-the-art methods. Code is available at https://github.com/mongko007/frn.

## 1 Introduction

Hyperspectral images (HSIs) contain more spectral bands (channels) than RGB images, enabling them to capture richer emission information that more accurately reflects the properties of objects. As a result, HSIs are commonly used in applications such as medical imaging [33, 35, 3], remote sensing [55, 46, 34, 19], material classification [25, 26], and object tracking [48, 31], etc.

Conventional hyperspectral imaging systems typically employ a single 1D or 2D sensor to scan the scene along the spatial or spectral dimension, capturing hyperspectral information through prolonged, repeated exposures. However, this approach is not well-suited for dynamic scenes. The coded aperture snapshot spectral imaging (CASSI) system takes advantage of the sparsity of spectral data, acquiring

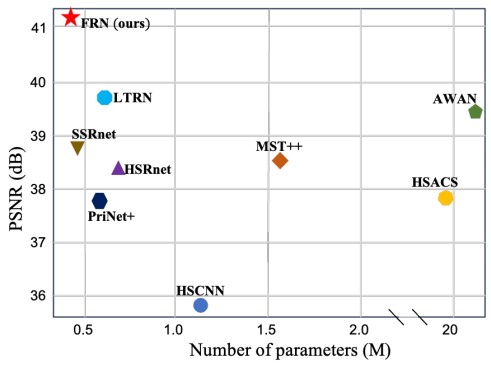

Figure 1: PSNR-Parameters comparisons of FRN and SOTA methods. FRN achieves outstanding HSI reconstruction performance with only a minimal number of parameters.

---

[*]Corresponding author.

39th Conference on Neural Information Processing Systems (NeurIPS 2025).

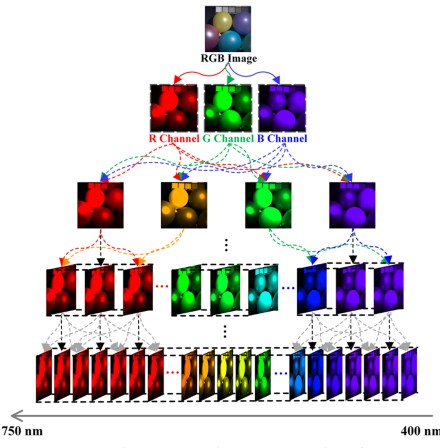

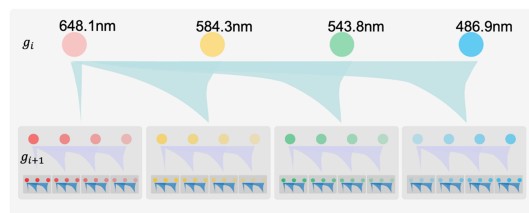

(b) The recursive invocation process of the atomic reconstruction module in FRN.

(a) A progressive spectral reconstruction framework.

Figure 2: Overview of the Fractal-Based Progressive Spectral Reconstruction Paradigm: (a) illustrates how FRN reconstructs images at specific wavelengths in a coarse-to-fine manner, transitioning from wide spectral bands to narrow ones across multiple levels. (b) demonstrates the structural self-similarity of the modules within each level.

compressed 2D measurements by modulating spectral signals at different wavelengths. The original HSIs are then reconstructed from these measurements using reconstruction algorithms [7, 9, 23, 24, 8, 36]. Despite its effectiveness, the high cost of CASSI devices has led researchers to explore more affordable alternatives. Given the widespread availability of RGB cameras, spectral reconstruction (SR) algorithms have been developed to recover HSIs from RGB input [10, 28, 20, 16, 52, 15].

The SR task is inherently ill-posed [15, 52]. Due to the variability in camera response functions, a single RGB image may correspond to multiple HSIs, complicating the accurate retrieval of hyperspectral information from limited RGB data. Traditional methods rely on statistical principles or hand-crafted sparse priors [2, 44]. However, the scarcity of paired RGB-HSI data limits the ability to fully capture and explore these prior assumptions. Convolutional neural networks (CNNs) have demonstrated strong performance in addressing various ill-posed problems, with some studies exploring their application to the SR task [15, 45, 29, 20]. Nevertheless, CNNs struggle to capture non-local spatial dependencies and model correlations between spectral bands. In contrast, transformers, owing to their multi-head self-attention mechanism, are better suited for handling long-range spatial dependencies and inter-band relationships, and have thus been widely applied to the SR task [10, 52, 11, 9, 8, 27]. However, such methods typically integrate spectral information from RGB images in a brute-force manner, leading to significant computational overhead and increased training complexity for neural networks.

Fractals are common patterns observed in neural networks [30]. Numerous studies have demonstrated the effectiveness of fractal or scale-invariant small-world network structures in the brain and its functional networks [4, 47, 6]. This inspires the idea that a larger network can be recursively constructed from smaller atomic modules. In addition to neural networks, fractal or near-fractal patterns are also commonly observed in natural data [30]. HSI data consists of sub-images from different bands, where sub-bands with slow variations in object emission characteristics can be treated as atomic cubes. Therefore, the SR task can be decomposed into the reconstruction of these sub-bands.

Based on the above analysis, we attempt to introduce the concept of fractals into the SR task. Fig. 1 shows the superior performance of our method. By employing the recursive principle, our method achieves superior HSI reconstruction performance while requiring only a minimal number of parameters. Fig. 2 (a) illustrates the progressive spectral reconstruction from wide to narrow bands. During the generation of a specific wavelength image, only spectral information from neighboring channels is used as cues for reconstruction, which aligns with the low-rank properties of HSI. Unlike integrating all hyperspectral information from an RGB image in a single step, this recursive generation approach mitigates the ill-posed problem by increasing the input and reducing the output at each

level. Fig. 2 (b) depicts our recursive generation framework, which exhibits self-similarity across levels by recursively invoking atomic reconstruction modules within atomic reconstruction modules.

A major challenge of recursively invoking the model is the substantial increase in the number of parameters and FLOPs. Visual Mamba (VMamba) models sequential dependencies by dividing an image into sequential blocks. A key advantage of VMamba is its linear computational complexity [57, 13, 42, 41, 43], which is particularly critical for handling high-dimensional hyperspectral features. Therefore, we design a state space model (SSM) with an adaptive band-aware mask (BAMamba) that filters out pixels with low spatial correlation before cross-scanning. This approach reduces computational cost while enabling the network to learn spatial sparsity effectively. In summary, our contributions are listed as follow:

1) We propose a fractal-based recursive spectral reconstruction network (FRN), to the best of our knowledge, it is the first attempt to introduce fractals to the SR task. FRN decomposes SR task into the reconstruction of sub-band images, establishing a new paradigm for spectral reconstruction.

2) FRN effectively leverages the low-rank characteristics of HSI while reducing the complexity of solving the ill-posed problem. Moreover, it exhibits self-similarity between atomic reconstruction modules across different levels.

3) We design a SSM with an adaptive band-aware mask (BAMamba), which reduces the computational cost by filtering out pixels with lower spatial correlation and forces the network to learn more compact pixel-wise inductive biases.

4) Through experiments conducted on different datasets, FRN demonstrates outstanding performance in both qualitative and quantitative metrics.

## 2 Related Work

### 2.1 Hyperspectral Image Reconstruction

Conventional hyperspectral imaging systems typically use spectrometers to scan scenes along the spatial or spectral dimensions. These scanners—such as pushbroom and whiskbroom types—have been widely applied in remote sensing, medical imaging, and environmental monitoring [5, 40]. However, they require long exposure times during scanning, rendering them unsuitable for dynamic scenes, and the imaging devices are too large to be easily portable. To overcome these limitations, snapshot compressive imaging (SCI) systems have been developed [12, 18, 32, 49]. These systems compress 3D HSI data into 2D measurements. The original HSIs are then reconstructed from these measurements using reconstruction algorithms. A representative example is the CASSI system. Despite their advantages, SCI systems are costly. As a more accessible alternative, many researchers have explored spectral reconstruction algorithms that aim to recover hyperspectral data from conventional RGB images [56, 53, 52, 10, 11], leveraging the widespread availability of RGB cameras.

### 2.2 Model-Based Methods

Model-based SR methods generally introduce prior knowledge to help the model reduce the difficulty of the ill-posed problem. Given the intrinsic low-rank nature of HSIs, sparse representation has become one of the most typically techniques for incorporating prior knowledge [38, 2, 44]. Some methods assume that the camera response function is known and learn a mapping from RGB images to hyperspectral reflectance [37, 27, 53]. However, this assumption is overly restrictive in practice. In addition, mathematical techniques such as singular value decomposition (SVD) [21] and Gaussian processes [1] are also employed to reconstruct HSIs. These methods place excessive reliance on priors, which constrains the model's representational ability and generalization capability.

### 2.3 Deep-Learning-Based Methods

Thanks to the powerful nonlinear fitting capabilities of CNNs, many researchers have applied them on SR. Convolutional blocks are used for spectral upsampling [53] or stacked layer by layer to enable the network to model more complex functions [45]. Considering the significant emission differences of objects across different spectral bands, efficient spatial-spectral attention mechanisms have been

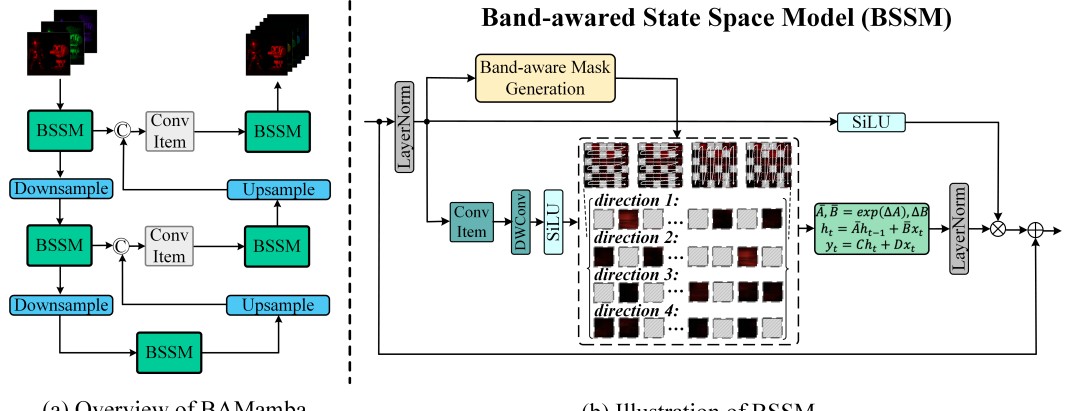

(a) Overview of BAMamba.       (b) Illustration of BSSM.

Figure 3: The details of BAMamba. BAMamba is a U-Net style network built with state space models equipped with band-aware masks (BSSM). BSSM introduces a band-aware spatial mask that adaptively perceives the reflectance of objects at specific wavelengths, suppressing interference from pixels with lower correlation.

designed to achieve more realistic reconstruction results [22, 28, 39]. To address the limitations of CNNs in capturing long-range spatial dependencies, a variety of Transformer-based SR methods have since emerged [10, 11, 52]. These methods learn inter-channel relationships by designing effective spectral-wise self-attention mechanisms. However, the above methods follow a one-shot reconstruction paradigm for HSI, overlooking the potential of recursive generation strategy to reduce the complexity of SR.

## 3 Method

### 3.1 Problem Formulation

The sensor of an RGB camera transfers the incident light to the R, G, and B channels through filters. This process can be regarded as an interaction between the camera response function (CRF) and the hyperspectral image

$$X\left(h,w,\lambda\right) = \int_{\lambda_{min}}^{\lambda_{max}} \phi\left(\lambda_i\right) \cdot Y\left(h,w,\lambda_i\right) \mathrm{d}\lambda, \tag{1}$$

where $X \in \mathbb{R}^{H \times W \times 3}$ represents the RGB image and $Y \in \mathbb{R}^{H \times W \times L}$ is the corresponding HSI. $Y\left(h,w,\lambda_i\right)$ denotes the spectral reflectance in the location$(h,w)$ at the wavelength of $\lambda_i$. $\phi\left(\lambda_i\right)$ represents the spectral response of the sensors at the wavelength of $\lambda_i$, and $[\lambda_{min}, \lambda_{max}]$ is the band range of $X\left(h,w,\lambda\right)$. The spectral information within the specified band range is integrated and stored in the channels of the RGB image. Eq. (1) can be rewritten in a discrete form

$$X\left(h,w,c\right) = \sum_{i=1}^{K} \phi\left(\lambda_i\right) \cdot Y\left(h,w,\lambda_i\right), \tag{2}$$

where $c \in [R, G, B]$ and $K$ is the number of channels within the corresponding band. Furthermore, Eq. (2) can be simplified as a matrix form

$$\mathbf{X} = \mathbf{Y}\Phi. \tag{3}$$

where $\mathbf{X} \in \mathbb{R}^{HW \times 3}$ denotes the vectorial representation of RGB image, and $\mathbf{Y} \in \mathbb{R}^{HW \times L}$ is the corresponding HSI with $L$ channels, $\Phi \in \mathbb{R}^{L \times 3}$ represents the CRF.

### 3.2 Spectral Reconstruction via Fractal Generator

Due to the self-similarity observed among local bands in HSI, we propose to construct a structurally self-similar SR network by following a recursive principle. We define a fractal generator $g_i$ as an

atomic module that generates next-level data $x_{i+1}$ from the previous-level result $x_i$: $x_{i+1} = g_i(x_i)$, as illustrated in Fig. 2 (b). Since the generator at each level can produce multiple outputs from a small amount of input, the fractal framework enables exponential growth of generated outputs with only a linear number of recursive levels [30], as shown in Fig. 2 (a). This property makes it particularly suitable for modeling high-dimensional HSI data using only a limited number of recursive levels. Specifically, we design an SSM with an adaptive band-aware mask (BAMamba) as the atomic generator, which will be described in Section 3.3.

The neural network learns the recursive principle from inter-band spectral correlations. For SR task, the objective is to learn the joint distribution of images at all wavelengths $p(y_{\lambda_1}, y_{\lambda_2}, \cdots, y_{\lambda_N})$. However, it is difficult to model the joint distribution in a single step. To address this, we adopt a progressive strategy, which can be viewed as a divide-and-conquer approach, which model the conditional distribution $p(y \mid x)$ at different generation levels. Assume that each atomic generator produces a data sequence of length $n$, and the number of channels in the HSI is $K$, let $K = n^m$, where $m = \log_n K$ is the number of recursive levels. The atomic generator at the first level divides the joint distribution $p(y_1, y_2, \cdots, y_K)$ into $n$ subsets, each containing $n^{m-1}$ variables. The joint distribution is decomposed

$$p(y_1, y_2, \cdots, y_K) = \prod_{i=1}^{n} p\left(y_{(i-1)\cdot n^{m-1}+1}, \cdots, y_{i\cdot n^{m-1}} \mid y_1, \cdots, y_{(i-1)\cdot n^{m-1}}\right). \quad (4)$$

Each conditional distribution is modeled by the atomic generator at the corresponding level. Fig. 2 illustrates the overall process. Through this typical divide-and-conquer strategy, FRN models the joint distribution over $K$ variables by employing $m$ levels of generators. The self-similarity of HSI along the spectral (channel) dimension, also known as its low-rank property, is effectively captured via the recursive principle, enabling a progressive approximation of the CRF $\Phi$ in Eq. (3).

### 3.3  Architecture of BAMamba

Visual Mamba (VMamba) exhibits linear computational complexity, enabling it especially well-suited for processing HSI data, which has a dimensionality much higher than that of RGB images. To address the computational burden introduced by recursive calls, we propose a VMamba-based sub-band generator that balances efficiency with performance, as shown in Fig. 3 (a).

The SSM employs a system of linear ordinary differential equations to connect inputs and outputs via intermediate hidden state representation. For a system with input signal $x(t) \in \mathbb{R}^L$, hidden state $h(t) \in \mathbb{C}^N$ and output response $y(t) \in \mathbb{R}^L$, the model can be formulated as

$$\begin{aligned} h'(t) &= \mathbf{A}h(t) + \mathbf{B}x(t), \\ y(t) &= \mathbf{C}h(t) + \mathbf{D}x(t), \end{aligned} \quad (5)$$

where $\mathbf{A} \in \mathbb{C}^{N \times N}$, $\mathbf{B}, \mathbf{C} \in \mathbb{C}^N$ and $\mathbf{D} \in \mathbb{C}^1$ are weighting parameters. Eq. (5) is typically discretized using a zero-order keeper (ZOH)

$$\begin{aligned} \overline{\mathbf{A}} &= \exp(\Delta\mathbf{A}), \\ \overline{\mathbf{B}} &= (\Delta\mathbf{A})^{-1}(\exp(\Delta\mathbf{A}) - I) \cdot \Delta\mathbf{B}, \end{aligned} \quad (6)$$

where $\Delta$ is a time scale parameter used to transform the continuous parameters $\mathbf{A}$, $\mathbf{B}$ into discrete parameters $\overline{\mathbf{A}}, \overline{\mathbf{B}}$. The discretized Eq. (5) can be written as

$$\begin{aligned} h_t &= \overline{\mathbf{A}}h_{t-1} + \overline{\mathbf{B}}x_t, \\ y_t &= \mathbf{C}h_t + \mathbf{D}x_t, \end{aligned} \quad (7)$$

Different objects may exhibit substantial differences in emissivity at the same wavelength, which implies that the energy intensity in HSI can undergo significant spatial variations. Furthermore, as illustrated in Fig. 4, objects exhibit varying degrees of distinction across different spectral bands.

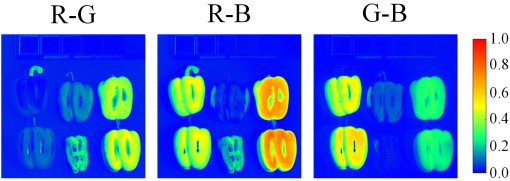

Figure 4: Residual maps across the R, G, and B channels from a CAVE dataset sample.

In SSMs, hidden states capture long-range dependencies by propagating historical information along the sequence. By accumulating and carrying previous data from earlier time steps, they enable

the model to retain past context and effectively establish a global receptive field. However, variations in the spatial distribution of spectral features across different bands can negatively impact the network's generalization ability. As shown in Fig. 4, significant differences may exist between the same (or different) objects across different (or the same) spectral bands. To mitigate the influence of accumulated band-specific information in hidden states, we attempt to suppress the interference from low-correlation regions by generating band-aware masks. According to the Eq. (6), $\Delta \mathbf{A}$ controls the impact of the current input sequence $x_t$ on the hidden states, with a positive correlation. Based on the value of the coefficients in $\Delta \mathbf{A}$, a band-aware mask $\mathbf{M}$ is generated during each sub-band generation

$$\mathbf{M} = \left\{ \begin{array}{ll} 1, & \Delta \mathbf{A} \geq \epsilon \\ 0, & \Delta \mathbf{A} < \epsilon \end{array} \right. , \tag{8}$$

$$y_t = (\mathbf{C} \odot \mathbf{M}) \, h_t, \tag{9}$$

where $\epsilon \in [0, \alpha]$, and $\alpha$ is a hyperparameter. According to Eq. (8) and Eq. (9), features in the hidden states associated with coefficient $\Delta \mathbf{A}$ that fall below a defined threshold are suppressed. As illustrated in Fig. 3 (b), some pixels with low spatial correlation (shaded areas) are filtered out. The overall working mechanism of the Band-awared SSM block (BSSM) is as follows

$$feat_{k+1} = \mathrm{CS} \left( \mathrm{LN} \left( feat_k \right) \right) \odot \mathrm{SiLU} \left( \mathrm{LN} \left( feat_k \right) \right) + feat_k. \tag{10}$$

where $feat$ denotes the backbone features and $\mathrm{CS} \left( \cdot \right)$ represents the band-aware scanning operation which employs the following operation sequence: $DWConv \rightarrow SiLU \rightarrow SSM \rightarrow LN$.

### 3.4 Loss Function

In this paper, we use $\mathcal{L}1$ loss to optimize the reconstructed HSI at the pixel level

$$\mathcal{L}_{rec} = \frac{1}{H \times W \times C} \sum_{i=0} \left| \widehat{Y} \left( i \right) - Y \left( i \right) \right|. \tag{11}$$

where $C$ is the number of channels in the reconstructed HSI, $\widehat{Y} \left( i \right)$ represents the predicted value for pixel $i$, and $Y \left( i \right)$ is its corresponding ground truth.

## 4 Empirical Results

### 4.1 Experimental Settings

**Dataset.** To validate the effectiveness of the proposed network, we conducted experiments on two datasets. The first dataset is the CAVE dataset [54] provided by Columbia University, which contains 32 HSIs. Each HSI consists of 31 spectral bands with a spectral interval of 10 nm, covering the spectral range from 400 nm to 700 nm. We randomly selected 20 HSIs for training, 6 HSIs for validation, and 6 HSIs for testing. The second dataset is the Harvard dataset [14] provided by Harvard University. It contains 50 HSIs covering both indoor and outdoor scenes. Each HSI consists of 31 spectral bands with a 10 nm interval, covering the spectral range from 420 nm to 720 nm. We randomly selected 30 HSIs for training, 10 HSIs for validation, and 10 HSIs for testing.

**Implementation Details.** We implemented our network on the PC with a single NVIDIA RTX 4090 GPU and built it in the PyTorch framework. In the training phase, the Adam optimizer [17] was used to optimize the model parameters. The initial learning rate was set to $4 \times 10^{-4}$ , and the learning rate was decayed using a cosine annealing schedule with a minimum value of $1 \times 10^{-6}$. The batch size was set to 32. We cropped $64 \times 64$ patches from 3D cubes and input them into the network. We set the number of recursive levels to 5, where each atomic generation module reconstructs a two-channel image. The threshold parameter $\alpha$ in Eq. (8) is empirically set to $0.5$.

### 4.2 Baseline Methods

We compared our FRN with seven SOTA spectral reconstruction methods: HSACS [29], SSRNet [16], HSRNet [20], AWAN [28], MST++ [10], LTRN [15], and MSFN [52].

## 4.3 Metrics

To evaluate the reconstruction quality of HSIs, we adopt four widely used IQA metrics: peak signal-to-noise ratio (PSNR), structural similarity index (SSIM) [51], root mean square error (RMSE), and universal image quality index (UIQI) [50]. PSNR quantifies reconstruction quality by computing the ratio of signal variance to noise, providing a measure sensitive to pixel-level errors. SSIM assesses the perceptual similarity between the reconstructed image and the ground truth by jointly considering luminance, contrast, and structural information. UIQI evaluates the consistency of pixel distributions by comparing the means and variances of the reconstruction and ground truth. Moreover, we compared the number of parameters of FRN with those methods.

Table 1: Performance comparison of different methods on CAVE and Harvard datasets. The best values are bolded. The up or down arrow indicates a higher or lower metric, corresponding to better performance.

| Method | Params(M) | CAVE | | | | Harvard | | | |
| --- | --- | --- | --- | --- | --- | --- | --- | --- | --- |
| | | PSNR↑ | RMSE↓ | UIQI↑ | SSIM↑ | PSNR↑ | RMSE↓ | UIQI↑ | SSIM↑ |
| HSACS | 19.74 | 37.9112 | 5.4099 | 0.8389 | 0.9765 | 42.1360 | 3.3203 | 0.8586 | 0.9762 |
| SSRNet | 0.39 | 38.6807 | 4.9824 | 0.8573 | 0.9794 | 42.1070 | 3.5370 | 0.8608 | 0.9760 |
| HSRNet | 0.77 | 38.4459 | 4.6511 | 0.8527 | 0.9801 | 41.6952 | 3.5459 | 0.8571 | 0.9747 |
| MST++ | 1.62 | 38.5511 | 4.6304 | 0.8731 | 0.9832 | 42.4756 | 3.2552 | 0.8623 | 0.9773 |
| AWAN | 21.36 | 39.4262 | 4.8245 | 0.8597 | 0.9798 | 42.2312 | 3.5034 | 0.8616 | 0.9769 |
| LTRN | 0.67 | 39.7349 | 4.3095 | 0.8702 | 0.9832 | 42.4953 | 3.3112 | 0.8632 | 0.9770 |
| MSFN | 2.48 | 39.8430 | 4.0372 | 0.8877 | 0.9860 | 42.6455 | 2.9916 | 0.8715 | 0.9771 |
| Ours | **0.30** | **41.0522** | **3.6243** | **0.9010** | **0.9900** | **42.8762** | **2.8933** | **0.8791** | **0.9774** |

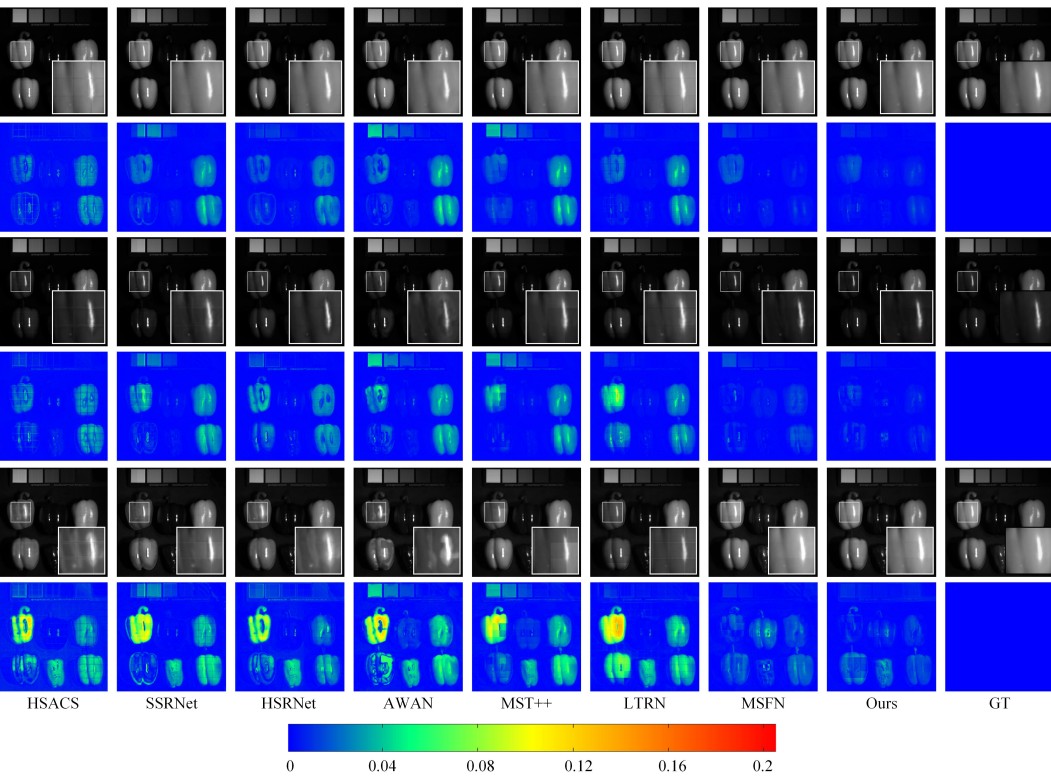

Figure 5: Comparison of the reconstruction results of different methods on one scene from the CAVE dataset, including seven SOTA methods and our FRN. We select three bands (20, 25, and 31) for visualization.

## 4.4 Performance Evaluation

**Numerical Results.** The quantitative results of different methods on the CAVE and Harvard datasets are presented in Tab. 1. Our method consistently achieves superior performance across all evaluation metrics. On the CAVE dataset, the average PSNR and SSIM of our method reach 41.05 dB and 0.99, respectively, outperforming the second-best results by 1.2 dB and 0.004. On the Harvard dataset, our method achieves a PSNR of 42.87 dB, surpassing the second-best results by 0.23 dB. Additionally, it can be observed that, compared to other methods, FRN achieves higher reconstruction quality while requiring the fewest model parameters.

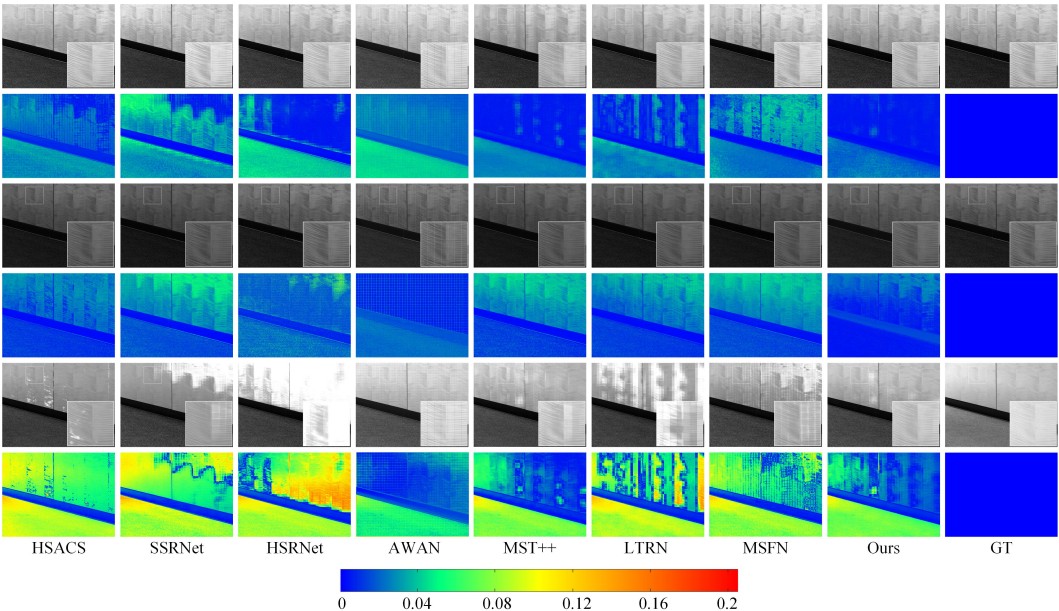

Figure 6: Comparison of the reconstruction results of different methods on one scene from the Harvard dataset, including seven SOTA methods and our FRN. We select three bands (10, 20, and 31) for visualization.

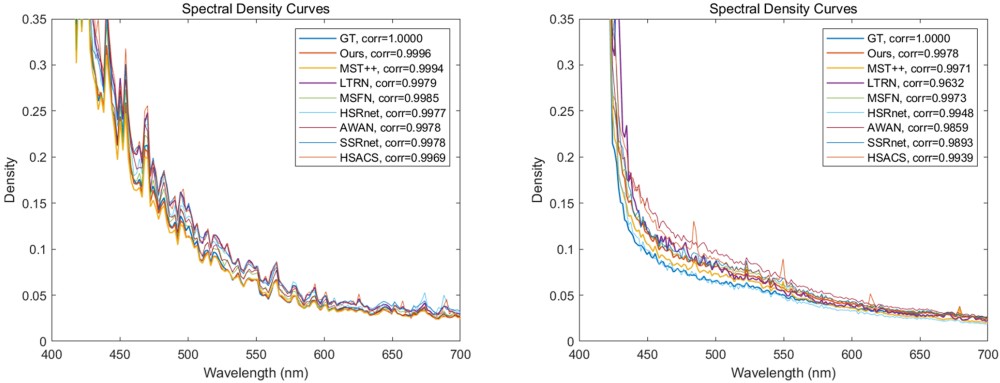

Figure 7: Comparison of the spectral curves among different methods on CAVE (left) and Harvard (right) datasets.

**Visual Results.** Fig. 5 and Fig. 6 show the reconstruction results of different methods on the CAVE and Harvard datasets, respectively. We select three channels from each scene for qualitative comparison. To provide a more intuitive comparison, we also present the residual maps between the predicted results and the ground truth. By zooming into local regions, it can be observed that our method reconstructs results that are closer to the ground truth in terms of spatial details and contrast. The corresponding spectral curve in Fig. 7 indicates FRN achieves higher spectral accuracy.

Table 2: Ablation studies of suppression threshold $\alpha$ on CAVE dataset. 'w/o' refers to the setting where the band-aware mask is not used (only with vanilla VMamba).

| Config | PSNR↑ | RMSE↓ | UIQI↑ | SSIM↑ |
|---|---|---|---|---|
| w/o | 39.7482 | 4.2998 | 0.8758 | 0.9840 |
| $\alpha = 0.2$ | 39.9022 | 4.1562 | 0.8820 | 0.9852 |
| $\alpha = 0.3$ | 40.3285 | 3.8683 | 0.8911 | 0.9866 |
| $\alpha = 0.5$ | **41.0522** | **3.6243** | **0.9010** | **0.9900** |
| $\alpha = 0.7$ | 40.1822 | 4.1022 | 0.8854 | 0.9860 |
| $\alpha = 0.8$ | 37.4822 | 5.8448 | 0.7822 | 0.9573 |

Table 3: Ablation studies of the number of recursive levels on CAVE dataset. 'w/o' means reconstruct the HSI from the RGB image in one step.

| Config | PSNR↑ | RMSE↓ | UIQI↑ | SSIM↑ |
|---|---|---|---|---|
| w/o | 39.8644 | 3.8683 | 0.8898 | 0.9883 |
| $M$=2 | 40.2210 | 3.7996 | 0.9001 | 0.9884 |
| $M$=3 | 40.6382 | 3.7057 | 0.9004 | 0.9886 |
| $M$=5 | **41.0522** | **3.6243** | **0.9010** | **0.9900** |

Table 4: Ablation studies of the number of reference spectral on CAVE dataset. 'w/o RGB' refers to excluding the RGB image from each reconstruction level.

| Config | PSNR↑ | RMSE↓ | UIQI↑ | SSIM↑ |
|---|---|---|---|---|
| w/o RGB | 38.6244 | 5.1320 | 0.8448 | 0.9773 |
| $S$=2 | 40.3981 | 3.8862 | 0.8962 | 0.9884 |
| $S$=3 | 40.8286 | 3.7900 | 0.8989 | 0.9886 |
| $S$=4 | **41.0522** | **3.6243** | **0.9010** | **0.9900** |
| $S$=5 | 40.8744 | 3.7458 | 0.8993 | 0.9888 |

## 4.5 Ablation Study

**Band-awared Mask.** We investigated the impact of the suppression threshold $\alpha$. Tab. 2 shows that setting $\alpha$ either too high or too low negatively affects the network performance. When the threshold is too low, the suppression effect is weakened, and low-correlation regions introduce redundant interference to feature learning. Conversely, an excessively high $\alpha$ causes the SSM to suppress informative features, resulting in information loss.

**Recursive Levels.** We evaluated the impact of the number of recursive levels $M$ on the reconstruction performance on CAVE dataset. By default, we set $M = 5$, which means that each BAMamba generated 2 new spectral channels based on the input from the previous level ($2^5 = 32$). In addition, we also attempted to reconstruct the HSI from the RGB image in one step. Tab. 3 shows that the reconstruction quality improves as the number of recursive levels increases. This is because each atomic generation step deals with fewer unknowns, thereby reducing the difficulty of solving the ill-posed problem.

**Spectral Cues.** We conducted ablation studies on the number of reference channels (wavelengths), denoted as $S$, fed into each atomic generator. By default, we input RGB images into each atomic generator to provide spectral priors. In Tab. 4, we found that setting $S = 4$ yields a relatively optimal performance. A smaller value of $S$ limits the amount of reference information available to the network, while a larger value may introduce noisy features. Another notable finding is that excluding the RGB image from each reconstruction level leads to a substantial decline in reconstruction quality. This degradation is attributed to the loss of structural and contrast priors inherently embedded in the RGB image, which are essential for accurate spectral learning.

## 5 Conclusion

In this paper, we propose a fractal-based recursive spectral reconstruction network (FRN). FRN establishes a new paradigm for spectral reconstruction by recursively invoking an atomic reconstruction module to progressively predict spectra from wide to narrow bands. By introducing the concept of fractals, FRN aligns with the low-rank nature of HSI data and exhibits structural self-similarity across different levels of the network. Furthermore, to alleviate the computational burden caused by the recursive design, we develop BAMamba, an atomic generation module based on SSM. Extensive experiments on multiple datasets demonstrate that FRN achieves outstanding performance in HSI reconstruction. Nevertheless, the recursive calling mechanism introduces significant computational overhead, which represents an important direction for future optimization.

# 6    Acknowledgments

This work was supported in part by the Dreams Foundation of Jianghuai Advance Technology Center project under Grant 2023-ZM01D002; ; in part by the National Natural Science Foundation of China under Grant 82172073, and Grant 62271430; and in part by the Open Fund of the National Key Laboratory of Infrared Detection Technologies.

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
