# OpenReview forum: "FRN: Fractal-Based Recursive Spectral Reconstruction Network"
_NeurIPS.cc/2025/Conference — NeurIPS 2025 poster_

### Official Review · Reviewer_opod · 2025-06-19

**Clarity:** 4
**Significance:** 4
**Originality:** 3
**Rating:** 6
**Confidence:** 5

**Summary:**

The author introduces the concept of fractals into the spectral reconstruction task, treating spectral reconstruction as a progressive process—either predicting from broad to narrow bands or adopting a coarse-to-fine approach to infer the next wavelength. This strategy reduces the overall task complexity and offers a novel perspective. Extensive experiments conducted on different datasets further demonstrate the effectiveness of the proposed approach.

**Questions:**

The author needs to provide a detailed explanation in Section 4.5 regarding how the reference channels were selected.

**Ethical Concerns:**

["NO or VERY MINOR ethics concerns only"]

**Final Justification:**

After reviewing the author’s responses to all the reviewers, I find that the author provides a more detailed explanation of the paper’s insight, so I decide to increase my score.

**Limitations:**

See Weaknesses.

**Quality:**

3

**Strengths And Weaknesses:**

Strengths：

1.The author is the first to introduce the mathematical concept of fractals into the spectral reconstruction task, formulating it as a progressive process. By constraining the generation of intermediate results, this approach effectively reduces the complexity of the problem and provides a fresh, innovative perspective.

2.The quantitative and visual experiments conducted by the author on different datasets demonstrate the effectiveness of the proposed method, while the ablation studies further confirm the validity and rationality of the core modules.

3.The overall description of the method in this paper is clear and easy to follow.

Weaknesses：

1.While the recursive approach is innovative, it may also introduce additional complexity, potentially limiting its applicability in real-time scenarios.

2.Section 4.1 lacks a description of the training epoch settings, which is crucial for ensuring reproducibility.

3.The paper lacks a detailed explanation of the reference channels discussed in Section 4.5, which may affect the clarity and understanding of the proposed method.

---

> ### Author Rebuttal · Authors · 2025-07-24
>
> Thank you for your review.
>
> 1. In fact, since the FRN requires multiple recursive calls to the atomic network, the GFLOPs increase accordingly. Taking this into consideration, we designed the lightweight BAMamba. The comparison of GFLOPs among different methods is as follows:
> Method | GFLOPs$\\downarrow$
> --------|--------
> HSACS | 82.18
> HSRNet | 3.00
> SSRNet | 1.58
> AWAN | 16.52
> MST++ | 1.46
> LTRN | $\\textbf{0.85}$
> MSFN | 2.04
> $\\textbf{FRN}$ | 1.45
>
> 2. The default setting for epochs is $300$.
>
> 3. At each stage, the input to $f(\\cdot)$ includes not only the RGB image but also the intermediate outputs from the previous stage, referred to as "Spectral Cues". In practice, we select several channels with the smallest Euclidean distances between all intermediate outputs from the previous stage and the current wavelength to be reconstructed by BAMamba, using them as additional cues. We found that the reconstruction results are optimal when the number of reference channels is $4$.

---

> > ### Comment · Reviewer_opod · 2025-08-05
> > **Official Comment by Reviewer opod**
> >
> > I thank the authors for providing such a detailed response in the rebuttal. After reading it carefully, I found that the authors addressed most of the issues that I raised in the initial phase. For instance, the additional GFlops of Method and detailed settings. To this end, I choose to keep my score.

---

### Official Review · Reviewer_5ryr · 2025-06-28

**Clarity:** 3
**Significance:** 2
**Originality:** 2
**Rating:** 3
**Confidence:** 4

**Summary:**

This paper introduces FRN, a network for reconstructing hyperspectral images (HSIs) from RGB images, addressing the limitations of existing methods with a unique approach based on fractal principles and progressive reconstruction. BAMamba, a band-aware state space model, further enhances efficiency and accuracy by filtering out irrelevant information.

**Questions:**

See weakness.

**Ethical Concerns:**

["NO or VERY MINOR ethics concerns only"]

**Final Justification:**

The authors did not directly address our main concerns in the reply.  The author's responses merely repeated what they had done, avoiding discussion of some core issues, including the difference between claimed contribution and practical implementation, and the lack of ablation experiments with other methods.

1. Although the authors claimed that this work is a fractal-based recursive method and emphasized that it is the first time to introduce the concept of fractals to the spectral reconstruction task, there is no real relationship between this work and fractal-based design. It's just a basic recursive network; the only connection is that they both use a recursive process.

2. The authors seem not to understand the difference between the self-ablation experiment and the comparative ablation experiment. The self-ablation experiment aims to demonstrate that incremental improvements are effective. This does not show any advantages of the proposed method over existing designs.

In summary, this work indeed has some innovation, but they are overclaimed and not solid. (And there have already been a large number of mamba-based works in spectral reconstruction fields, which are not referred to.)  The experiments also fail to demonstrate what real improvements this design has over other existing methods, rather than simple baselines.

**Limitations:**

yes

**Paper Formatting Concerns:**

n

**Quality:**

2

**Strengths And Weaknesses:**

Strengths：

1. Progressive reconstruction is a feasible idea and would have been used in scenarios with limited resources.

2. FRN consistently achieves superior performance compared to exsiting methods in both quantitative and qualitative evaluations, reconstructing HSIs with higher accuracy and detail.

3. The recursive design and BAMamba module effectively reduce the computational burden of the reconstruction process, making FRN more efficient and suitable for handling high-dimensional HSI data.

Weaknesses:

1. Fractals are generally adopted in two-dimensional [1] and above spaces. In the one-dimensional space of the spectrum, the fractal does not more sense than simple linear interpolation.

2. The Mamba model has been widely applied in the field of spectral reconstruction, and the related methods[2-7] are not discussed in this paper. From the current manuscript, we cannot find contributions beyond the previous work.

3. The core contributions of this work include two parts: the fractal reconstruction framework and the Mamba-based network module. The former is a progressive reconstruction network, and it is hard to say that it truly surpasses the previous framework design. The latter introduced the Mamba network and made no significant contribution compared with the previous spectral reconstruction work based on Mamba. Therefore, the contribution of this work is very limited.

[1] Fractal generative models, arvix 2025.

[2] MambaHSI: Spatial-spectral mamba for hyperspectral image classification, TRGS 2024.

[3] Spectral-spatial mamba for hyperspectral image classification, arvix 2024.

[4] Dual hyperspectral mamba for efficient spectral compressive imaging, arvix 2024.

[5] VmambaSCI: Dynamic Deep Unfolding Network with Mamba for Compressive Spectral Imaging, MM 2024.

[6] MambaSCI: Efficient Mamba-UNet for Quad-Bayer Patterned Video Snapshot Compressive Imaging, NeurIPS 2024.

[7] Detail Matters: Mamba-Inspired Joint Unfolding Network for Snapshot Spectral Compressive Imaging, AAAI 2025.

---

> ### Author Rebuttal · Authors · 2025-07-25
>
> Thank you for your review.
> 1. In fact, the authors of [1] did not state that "fractals are generally adopted in two-dimensional and above spaces." The core of fractals lies in the "$\\textbf{self-similarity}$" inherent in both $\\textbf{the data and the model}$. For hyperspectral image (HSI) data, the strong correlation between adjacent spectral bands precisely indicates the presence of self-similarity along the spectral dimension. The HSI reconstruction task is essentially an ill-posed problem of generating many more bands (e.g., 31-channel HSI) from a few bands (e.g., RGB 3-channel image). This task can be decomposed into the reconstruction of atomic bands, and the networks involved in each sub-process also possess structural self-similarity. Therefore, the fractal concept is very well matched to the task addressed in this paper.
>
> 2. These Mamba-based spectral reconstruction methods have not explored the impact of SSM on spectral reconstruction from the perspective of $\\textbf{redundant information removal}$. Redundant information among spectral bands is one of the key factors hindering reconstruction performance. The band-aware mask introduced in BAMamba is a unique design in this paper aimed at addressing this issue. This scheme enables Mamba to learn more compact spectral and spatial correlations. Furthermore, existing studies typically stack multiple SSM layers to improve network performance, which weakens the inherent advantage of SSM’s low computational complexity. In contrast, the introduction of the mask reduces the involvement of irrelevant information in computations, further enhancing $\\textbf{inference efficiency}$. Experimental results also validate the effectiveness of this scheme in both reconstruction quality and efficiency. Therefore, BAMamba’s design demonstrates clear innovation over previous approaches from the perspective of $\\textbf{task understanding}$.
>
> 3. The progressively designed spectral reconstruction network based on fractal principles represents $\\textbf{a novel paradigm}$ for spectral reconstruction. It fundamentally reduces the difficulty of solving the ill-posed problem, which was $\\textbf{not addressed by previous methods}$. Experimentally, our method significantly outperforms prior approaches, and we believe it will provide valuable references for the community. On the other hand, by leveraging the physical or statistical characteristics of HSI data, this paper employs SSM as a backbone component to reduce the resource consumption required for processing high-dimensional data. Additionally, by introducing a mask, redundant information interference is partially eliminated, further improving the model’s inference efficiency and mitigating the negative effects caused by recursive calls.
>
> [1] Fractal generative models, arvix 2025.

---

> > ### Comment · Reviewer_5ryr · 2025-08-02
> >
> > Thanks for the reply.
> >
> > For the first point, I think the conception of "$\textbf{self-similarity}$" does not apply to the spectral dimension, and there is no strong correlation between distant spectral bands. For the current method, "$\textbf{local similarity}$" is a more accurate conception. Secondly, true fractal methods utilize self-similarity, but this does not mean that using self-similarity is fract-based, such as spectral attention. Finally, the current method is merely based on recursion and has no substantial connection with fractals.
> >
> > For the second point, band awareness does seem original for the mamba methods, but it is not uncommon in other types of methods.
> >
> > For the third point, I think this framework is indeed relatively original, but the authors have not "$\textbf{experimentally}$" proved that the framework actually has any superiority. The recursive approach requires multiple reconstructions and does not have a significant advantage over a single end-to-end reconstruction.
> >
> > I have also noticed that the experiments in the current paper are very weak. The authors do not compare the designs of $\textbf{any other mamba methods or any other frameworks}$. Moreover, the current method is not implemented on $\textbf{commonly used real datasets}$, making it difficult to verify its effectiveness in practical applications.
> >
> > In addition, the authors adopt a completely different experimental setup from the previous methods and do not compare the proposed method under the commonly used setup, which is questionable. The current work chooses to conduct training and testing on the same dataset and does not adopt cross-validation, which is inappropriate and cannot verify the generalization of the approach.

---

> > > ### Author Response · Authors · 2025-08-02
> > >
> > > Thank you for your comments.
> > >
> > > 1. In fact, I $\\textbf{mentioned}$ in the rebuttal: "1. For hyperspectral image (HSI) data, the strong correlation between $\\textbf{adjacent}$ spectral bands precisely indicates the presence of self-similarity along the spectral dimension." This is consistent with your description of "local similarity", and that’s why the relationship between $\\textbf{adjacent bands}$ is regarded as a form of "fractal" in this paper. Moreover, spectral attention mechanisms and fractals are entirely different concepts — the former applies linear or nonlinear weighting to feature representations, while the latter is a "local generation strategy".
> > >
> > > 2. The band-aware mechanism in this paper is $\\textbf{specifically designed}$ for the HSI reconstruction task. Its purpose is to reduce the computational burden of the SSM, while suppressing interference from irrelevant regions and preventing network overfitting.
> > >
> > > 3. In fact, the experiments in the paper addressed your concern. Regarding the contributions of the fractal recursion, we provided a "$\\textbf{w/o}$" configuration in $\\textbf{Table 3}$, which referred to reconstructing the HSI from the RGB image in a single step (without using fractal recursion). It was observed that all metrics dropped significantly on the CAVE dataset (with PSNR decreasing by $\\textbf{1.18}$ dB), demonstrating that the proposed framework brought a notable performance improvement.
> > >
> > > 4. The publicly available CAVE and Harvard datasets are $\\textbf{currently the two most commonly used benchmarks [1-2]}$. We are not sure what you specifically mean by "commonly used real datasets." Moreover, the performance of FRN has already surpassed existing SOTA methods.
> > >
> > > 5. Regarding the experimental setup, our settings follow $\\textbf{the latest}$ method [1], and all comparison methods use $\\textbf{the same configuration}$, so $\\textbf{there is no unfairness}$. Moreover, we did not train and test on the same dataset — we trained BAMamba separately on the two datasets.
> > >
> > > [1] Spectral super-resolution via deep low-rank tensor representation. Dian R, Liu Y, Li S, et al. IEEE Transactions on Neural Networks and Learning Systems, 2024.
> > >
> > > [2] Spectral super-resolution via model-guided cross-fusion network. Dian R, Shan T, He W, et al. IEEE Transactions on Neural Networks and Learning Systems, 2023.

---

> > > > ### Comment · Reviewer_5ryr · 2025-08-03
> > > >
> > > > Thanks for the reply.
> > > >
> > > > 1. The core of this framework is recursion, which can take advantage of local similarity. This is fine, but it cannot be called a fractal-based approach. There is no fractal-related design or content in the current method. Essentially, it is just basic recursion.
> > > >
> > > > 2. Theoretically, the mask can serve to determine whether to perform a calculation, thereby reducing the computational burden. However, this requires the design of the underlying architecture, but we did not see any related content in this article. Although the authors may think that masking can reduce the computation, in fact, the masked channels still participate in the computation, but have no impact on the final result. That is to say, the computational burden has not improved.
> > > >
> > > > 3. I recommend that authors review the experimental designs of previous papers such as MST[1] and DAUSHT[3]. Self-ablation experiments are necessary, but this does not mean that comparisons with other methods are not required. Otherwise, most modules can bring about performance improvements, which is meaningless.
> > > >
> > > > 4. The commonly used datasets are fine, but it is questionable that the experimental results under common configurations are not provided. It is easier to obtain a desired result by freely designing the datasets and making comparisons by oneself. Moreover, real datasets are necessary to demonstrate the performance of the method in actual tasks.
> > > >
> > > > 5. I recommend that the authors follow the experimental setup of the current mainstream methods [1-6], which I didn't notice before. I found that the author did not compare multiple solid works [1-6] in snapshot spectral imaging, which is very unreasonable. Moreover, the number of parameters given in the author's experimental table does not match the original paper.
> > > >
> > > > [1] Mask-guided Spectral-wise Transformer for Efficient Hyperspectral Image Reconstruction, in CVPR 2022.
> > > >
> > > > [2] Coarse-to-Fine Sparse Transformer for Hyperspectral Image Reconstruction, in ECCV 2022.
> > > >
> > > > [3] Degradation-Aware Unfolding Half-Shuffle Transformer for Spectral Compressive Imaging, in MeurIPS 2022.
> > > >
> > > > [4] Residual Degradation Learning Unfolding Framework with Mixing Priors across Spectral and Spatial for Compressive Spectral Imaging, in CVPR 2023.
> > > >
> > > > [5] Dual Prior Unfolding for Snapshot Compressive Imaging, in CVPR 2024.
> > > >
> > > > [6] Latent Diffusion Prior Enhanced Deep Unfolding for Snapshot Spectral, Compressive Imaging, in ECCV 2024.

---

> > > > > ### Author Response · Authors · 2025-08-03
> > > > >
> > > > > Thank you for your comments.
> > > > >
> > > > > 1. The fundamental idea of fractals is based on recursion. The key lies in how to introduce the concept of "fractals" into a specific task.
> > > > >
> > > > > 2. In practice, we controlled the input sequence size of the SSM by setting hyperparameter (masked pixels did not participate in the computation).
> > > > >
> > > > > 3-5. The methods you listed are not within the scope of this paper. $\\textbf{These methods are not comparable.}$ Mainstream hyperspectral reconstruction approaches can be divided into two categories: one is reconstruction based on RGB images (which is the focus of this paper), and the other is Snapshot Compressive Imaging (covered by the papers you mentioned). These are two entirely different tasks, with distinct input data and information processing pipelines. I believe this is one of the reasons why you have misunderstood many aspects of the experimental section. Furthermore, the only publicly available RGB-HSI paired datasets are CAVE, Harvard, and NTIRE.

---

> > > > > > ### Comment · Reviewer_5ryr · 2025-08-07
> > > > > >
> > > > > > Thanks for the reply.
> > > > > >
> > > > > > The methods of [1-6] indeed do not fall into this category. However, the two main concerns of the current work--- the actual design has no relationship with the claimed fractal-based contribution, and the limited experimental design---have not been addressed.

---

> > > > > > > ### Author Response · Authors · 2025-08-08
> > > > > > >
> > > > > > > Thank you for your comments.
> > > > > > >
> > > > > > > 1. The fractal concept based on recursive design is $\\textbf{reasonable}$ [1].
> > > > > > >
> > > > > > > 2. Our Table 1 compares the proposed method with $\\textbf{7 state-of-the-art approaches}$ (including two latest methods published in $\\textbf{2024}$), $\\textbf{demonstrating its overwhelming superiority}$.
> > > > > > >
> > > > > > > 3. Table 2, through ablation studies, verifies the effectiveness of the $\\textbf{mask mechanism in SSM}$; Table 3 demonstrates the effectiveness of the $\\textbf{recursive mechanism (including recursive depth)}$; and Table 4 shows the impact of $\\textbf{the number of channels used as spectral cues}$ during the atomic generation process.
> > > > > > >
> > > > > > > [1] Fractal generative models, arvix 2025.

---

### Official Review · Reviewer_fsWJ · 2025-07-01

**Clarity:** 3
**Significance:** 4
**Originality:** 4
**Rating:** 6
**Confidence:** 5

**Summary:**

The authors creatively introduce the mathematical concept of fractals into spectral reconstruction by recursively reconstructing atomic spectral cubes, thereby reducing the complexity of the task. Furthermore, to alleviate the computational burden introduced by recursion, the authors adopt an efficient design based on the Mamba architecture. Experimental results validate the effectiveness of the proposed approach.

**Questions:**

The author needs to provide a more detailed explanation of how M is selected and how the final number of generated channels is determined.

**Ethical Concerns:**

["NO or VERY MINOR ethics concerns only"]

**Final Justification:**

The authors have responded clearly and thoroughly to my concerns. They provided detailed clarification on the recursive generation process, including how the hyperparameter M influences the final number of spectral channels. The additional experimental results further support their design choices. Given the solid empirical results, I believe the contribution is stronger than I initially judged.

**Limitations:**

See Weaknesses.

**Quality:**

3

**Strengths And Weaknesses:**

Strengths:
[1] The author skillfully integrates the concept of fractals into spectral reconstruction and effectively leverages the Mamba network to alleviate the computational burden associated with multiple recursive calls.
[2] Decomposing the reconstruction of the entire HSI into the reconstruction of multiple atomic cubes, while progressively providing more information (inputs) for subsequent reconstructions and simultaneously reducing the outputs (unknowns), is an interesting idea.
[3] Experimental results confirm the effectiveness of FRN.
[4] The paper is clearly written and well composed.

Weaknesses:
[1] The description of the hyperparameter M in Section 4.5 is unclear, and it is also not explained how different values of M ultimately generate a HSI with a fixed number of channels.
[2] Section 4.5 lacks further experiments on the Recursive Levels, such as evaluating performance at M=6, to better justify the choice of this hyperparameter.
[3] The paper lacks a detailed explanation of the reference channels mentioned in Section 4.5.

---

> ### Author Rebuttal · Authors · 2025-07-24
>
> Thank you for your review.
>
> 1. We will include an example to more concretely illustrate how the recursive levels M (in Section 4.5) influence the hyperspectral image generation process. As shown in Figure.2 (a), in this paper, we input a 3-channel RGB image into the FRN and generate a $K=31$-channel hyperspectral image. However, the generation process is not an end-to-end operation in the traditional sense. Instead, it requires $M$ stages to complete, with each stage employing the same U-Net style BAMamba model as the atomic reconstruction network. Assuming the default setting of $M=5$, the output size of BAMamba at each stage is fixed to $2 \\times H \\times W,$ that is, $n=2$ spectral channels (where $M=\\log_{n}{K}$). The complete process is as follows:
>
> Stage 1 ($1$-call, ($n \\times 1=2$)-band generation):
>
> $f(x_r, x_g, x_b) = (x_1, x_2)$;
>
>
> Stage 2 ($2$-call, ($n \\times 2=4$)-band generation):
>
> $f(x_r, x_g, x_b, x_1) = (x_1, x_2)$, $f(x_r, x_g, x_b, x_2) = (x_3, x_4)$;
>
>
> Stage 3 ($4$-call, ($n \\times 4=8$)-band generation):
>
> $f(x_r, x_g, x_b, x_1) = (x_1, x_2)$, $f(x_r, x_g, x_b, x_2) = (x_3, x_4)$, $f(x_r, x_g, x_b, x_3) = (x_5, x_6)$, $f(x_r, x_g, x_b, x_4) = (x_7, x_8)$;
>
>
> Stage 4 ($8$-call, ($n \\times 8=16$)-band generation):
>
> $f(x_r, x_g, x_b, x_1) = (x_1, x_2)$, $f(x_r, x_g, x_b, x_2) = (x_3, x_4)$, $f(x_r, x_g, x_b, x_3) = (x_5, x_6)$, $f(x_r, x_g, x_b, x_4) = (x_7, x_8)$, $f(x_r, x_g, x_b, x_5) = (x_9, x_{10})$, $f(x_r, x_g, x_b, x_6) = (x_{11}, x_{12})$, $f(x_r, x_g, x_b, x_7) = (x_{13}, x_{14})$, $f(x_r, x_g, x_b, x_8) = (x_{15}, x_{16})$;
>
>
> Stage 5 ($16$-call, ($n \\times 16=32$)-band generation):
>
> $f(x_r, x_g, x_b, x_1) = (x_1, x_2)$, $f(x_r, x_g, x_b, x_2) = (x_3, x_4)$, $\\cdots$, $f(x_r, x_g, x_b, x_{15}) = (x_{29}, x_{30})$, $f(x_r, x_g, x_b, x_{16}) = (x_{31}, x_{32})$.
>
>
> The value of $M$ and the number of spectral channels in the HSI must satisfy the condition that $n^M$ is closest to $K$. This ensures that the FRN can generate a sufficient number of spectral bands without introducing excessive redundant outputs. For example, when $M=5$, the FRN ultimately produces $32$ channels ($>31$), and the extra final channel will be removed.
>
> 2. We further provide experimental results for the case of $M=6$ and $n=2$, where the FRN outputs $2^6=64$ channels. These were mapped to $32$ channels by computing the channel-wise mean, and the final extra channel was removed. The evaluation metrics are as follows:
> Config | PSNR$\\uparrow$ | RMSE$\\downarrow$ | UIQI$\\uparrow$ | SSIM$\\uparrow$
> --------|--------|--------|--------|--------
> $\\textbf{M=5}$ | $\\textbf{41.0522}$ | $\\textbf{3.6243}$ | $\\textbf{0.9010}$ | $\\textbf{0.9900}$
> $M=6$ | 41.0233 | 3.6441 | 0.9005 | 0.9887
>
> Deeper recursion introduced more redundant information into the reconstruction output, leading to a decline in reconstruction performance.
>
> 3. In each stage, the input to $f(\\cdot)$ includes not only the RGB image but also the intermediate outputs from the previous stage, referred to as "Spectral Cues" (as mentioned in Section 4.5). In practice, instead of simply adding one intermediate result from the previous stage as an additional condition to $f(\\cdot)$, we compute the Euclidean distances between all intermediate outputs of the previous stage and the current wavelength to be reconstructed by BAMamba. Then, we select the closest channels as additional cues. We found that the reconstruction achieves the best performance when the number of reference channels is set to $4$.
>
> 4. The values of $M$ and $n$ are determined such that $n^M$ approximates $31$ ($K$) as closely as possible.

---

> > ### Comment · Reviewer_fsWJ · 2025-08-05
> >
> > Thank you for the clarification. The authors have addressed my concern appropriately. I will keep my original score.

---

### Official Review · Reviewer_XWjn · 2025-07-01

**Clarity:** 3
**Significance:** 3
**Originality:** 3
**Rating:** 4
**Confidence:** 3

**Summary:**

This paper introduces FRN, a novel framework for reconstructing hyperspectral images (HSIs) from RGB inputs. Drawing inspiration from fractal geometry, FRN decomposes the spectral reconstruction task into a recursive hierarchy of atomic sub-band generators. This design progressively models spectral correlations from coarse to fine granularity, aligning with the self-similar and low-rank nature of hyperspectral data. Complementing this, the authors propose BAMamba, a state-space model enhanced with adaptive, band-aware spatial masking, which selectively suppresses spatially uninformative regions. Extensive experiments on CAVE and Harvard datasets demonstrate that FRN significantly outperforms state-of-the-art methods in both accuracy and parameter efficiency.

**Questions:**

1. In Table 3, deeper recursion levels appear to yield better performance. Is there a trade-off point where performance saturates or degrades due to over-recursion? Reporting trends beyond 5 levels or plotting performance vs. depth could offer insight into this trade-off.
2. Since the architecture is jointly inspired by fractal generative principles and integrates Mamba-style state-space modeling, it would be insightful to include an ablation that disentangles the contributions from the fractal recursion and from BAMamba. How much gain is attributable to each component?
3. Given the novelty and competitive performance, open sourcing would benefit the research community and increase the impact of the work.

**Ethical Concerns:**

["NO or VERY MINOR ethics concerns only"]

**Final Justification:**

The authors have addressed my concerns raised during the review stage. I consider this to be an above-average paper and therefore maintain my original rating.

**Limitations:**

Yes. The authors have clearly articulated the limitations of their work and have thoughtfully considered potential societal risks.

**Quality:**

3

**Strengths And Weaknesses:**

## Strengths
1. The use of fractal-based recursion to progressively reconstruct spectral bands introduces a new paradigm in hyperspectral image reconstruction, well-aligned with the hierarchical structure and redundancy in HSI data.
2. The BAMamba module effectively leverages the spatial sparsity of informative regions, reducing unnecessary information and improving generalization.
3. The proposed method achieves strong performance on standard datasets, outperforming prior works with notably fewer parameters, showcasing both effectiveness and efficiency.

## Weaknesses
1. While the model reduces parameter count and theoretical complexity, the recursive nature of inference introduces sequential dependencies that may hinder real-time performance, especially for high-resolution HSI reconstruction. Notably, the paper does not report any runtime or latency metrics, which would be essential for evaluating deployment feasibility.
2. The ablation studies, though helpful, do not fully disentangle the relative contributions of the fractal recursion and BAMamba (see following Additional Feedback). A dedicated component-wise analysis would help clarify their individual and joint effects on performance.
3. From the experimental setup, it appears that a separate model is trained for each dataset. The authors are encouraged to report or discuss the training cost.

---

> ### Author Rebuttal · Authors · 2025-07-26
>
> Thank you for your review. We will reply to your questions in order.
> 1. To address your concerns regarding the inference efficiency of FRN, we further provide a comparison of the inference time and memory cost required by each method for processing a single image (Taking a single sample from the CAVE dataset as an example.):
>
> Method | Time(S)$\\downarrow$ | Mem.(G)$\\downarrow$
> -------|-------|-------
> HSACS | 2.39 | 0.208
> HSRNet | 0.94 | 0.031
> SSRNet | 1.03 | 0.016
> AWAN | $\textbf{0.89}$ | 0.413
> MST++ | 1.08 | 0.028
> LTRN | 1.94 | 0.015
> MSFN | 3.69 | 0.069
> $\\textbf{FRN}$ | 0.92 | $\textbf{0.010}$
>
> It can be observed that FRN has the second fastest inference speed after AWAN and the lowest memory cost.
>
> 2. Thank you for your constructive comments. Regarding the contributions of the fractal recursion, we provide a 'w/o' configuration in Table 3, which means reconstructing the HSI from the RGB image in one step (without using fractal recursion). It can be observed that all metrics drop significantly on the CAVE dataset (with PSNR decreasing by $1.18$ dB).
>
> As for your suggestion on conducting a component-wise analysis of BAMamba, we attempted to replace the atomic generator with MSFN. The results on the CAVE dataset are as follows:
>
> Config | PSNR$\\uparrow$ | RMSE$\\downarrow$ | UIQI$\\uparrow$ | SSIM$\\uparrow$
> -------|-------|-------|-------|-------
> MSFN | 40.0611 | 3.9624 | 0.8890 | 0.9866
> $\\textbf{BAMamba}$ | $\\textbf{41.0522}$ | $\\textbf{3.6243}$ | $\\textbf{0.9010}$ | $\\textbf{0.9900}$
>
> It can be found that BAMamba outperforms MSFN, achieving a $0.99$ dB higher PSNR on the CAVE dataset. When comparing the contributions of fractal recursion and BAMamba in a component-wise manner, we observe that the former contributes more significantly. We attribute this to the fact that fractal recursion serves as a general strategy that fundamentally reduces the difficulty of solving the ill-posed problem. In contrast, the latter (a deep neural network for HSI reconstruction) may be more prone to overfitting due to potential limitations in its design.
>
> 3. We trained BAMamba separately on the two datasets, and all experiments were conducted on a single NVIDIA RTX 4090 GPU. The specific training details are provided in Section 4.1. The default setting for epochs is $300$. The training time on the CAVE dataset was $1.58$ hours with a memory cost of $12.8$ GB, while the training time on the Harvard dataset was $13$ hours with a memory cost of $12.8$ GB.
>
> 4. In fact, we later identified this experimental gap and further conducted experiments by setting the recursion levels to $M=6$ and $n=2$. The results on the CAVE dataset are as follows:
> Config | PSNR$\\uparrow$ | RMSE$\\downarrow$ | UIQI$\\uparrow$ | SSIM$\\uparrow$
> --------|--------|--------|--------|--------
> $\\textbf{M=5}$ | $\\textbf{41.0522}$ | $\\textbf{3.6243}$ | $\\textbf{0.9010}$ | $\\textbf{0.9900}$
> $M=6$ | 41.0233 | 3.6441 | 0.9005 | 0.9887
>
> Deeper recursion introduced more redundant information into the reconstruction output, leading to a decline in reconstruction performance.
>
> 5. Same as Question 2.
>
> 6. We will release our code on GitHub.

---

> > ### Comment · Reviewer_XWjn · 2025-08-05
> >
> > Thanks for the authors' feedback, which mainly addresses my concerns. I will keep my score.

---

> > > ### Author Response · Authors · 2025-08-05
> > >
> > > Thanks for your reply :)

---

### Note · Authors · 2025-08-11

Dear AC and Reviewers,

Thank you for your efforts. In this work, we introduce the concept of "fractals" into spectral reconstruction, establishing a completely new paradigm for this task. The strategy of recursively generating atomic bands helps to some extent reduce the difficulty of solving the inherently ill-posed problem of spectral reconstruction. The experimental results also fully demonstrate the effectiveness of our method.

$\textbf{All reviewers ultimately acknowledged the novelty of our method.}$ During the rebuttal and discussion period, their questions mainly focused on the experimental setup details and additional explanations of the module design. We provided responses to each of these concerns, and all reviewers indicated that their concerns had been addressed. However, one reviewer still had doubts regarding the overall architectural design.

It is worth noting that the experimental section of our paper, through $\textbf{comparisons with SOTA methods and ablation studies}$, has demonstrated both the $\textbf{effectiveness and the necessity}$ of each core component of the network, including but not limited to the recursive generation scheme and the masking mechanism in the SSM.

$\textbf{We sincerely hope that the AC will take our rebuttal process fully into consideration when making the final decision.}$

---

### Decision · Program_Chairs · 2025-09-17

**Decision:**

Accept (poster)

**Comment:**

This paper introduces FRN, a fractal-inspired recursive network for hyperspectral image reconstruction, complemented by a band-aware Mamba module (BAMamba) to enhance efficiency and accuracy. The key scientific claim is that modeling spectral correlations in a coarse-to-fine recursive manner aligns well with the self-similar, low-rank nature of hyperspectral data, while BAMamba selectively suppresses uninformative regions to improve generalization. Strengths highlighted by reviewers include the novelty of fractal-based recursion in HSI reconstruction, solid empirical results surpassing prior methods, parameter efficiency, and a clear presentation. Weaknesses include limited runtime analysis, incomplete ablations disentangling recursion and BAMamba, and insufficient discussion of related Mamba-based methods. During rebuttal, the authors clarified the recursive design, explained hyperparameter settings (e.g., M), and provided additional analysis on spectral channel generation, which satisfactorily addressed most concerns. Despite some remaining questions on broader comparisons, the overall contribution is innovative, impactful, and well supported by experiments.